# *Vibrio parahaemolyticus* from Migratory Birds in China Carries an Extra Copy of tRNA-Gly and Plasmid-Mediated Quinolone Resistance Gene *qnrD*

Lin Zheng,[a,c] Chao Yang,[b,d] Ping Chen,[c] Lingwei Zhu,[a] Huiqi Wen,[b] Mingwei Liu,[a,c] Jiayao Guan,[a] Gejin Lu,[a] Jie Jing,[a] Shiwen Sun,[a] Ying Wang,[a] Yajun Song,[b] Ruifu Yang,[b] Xianglilan Zhang,[b] Yujun Cui,[b] Xuejun Guo[a]

aChangchun Veterinary Research Institute, Chinese Academy of Agricultural Sciences/Key laboratory of Jilin Province for Zoonosis Prevention and Control, Changchun, China
bState Key Laboratory of Pathogen and Biosecurity, Beijing Institute of Microbiology and Epidemiology, Beijing, China
cSchool of Food and Engineering, Jilin Agricultural University, Changchun, China
dThe Center for Microbes, Development and Health, CAS Key Laboratory of Molecular Virology and Immunology, Institut Pasteur of Shanghai, Chinese Academy of Sciences, Shanghai, China

Lin Zheng and Chao Yang contributed equally to this work. Author order was determined by the corresponding author after negotiation.

**ABSTRACT** *Vibrio parahaemolyticus* is a marine bacterium coming from estuarine environments, where the migratory birds can easily be colonized by *V. parahaemolyticus*. Migratory birds may be important reservoirs of *V. parahaemolyticus* by growth and re-entry into the environment. To further explore the spreading mechanism of *V. parahaemolyticus* among marine life, human beings, and migratory birds, we aimed to investigate the characteristics of the genetic diversity, antimicrobial resistance, virulence genes, and a potentially informative gene marker of *V. parahaemolyticus* isolated from migratory birds in China. This study recovered 124 (14.55%) *V. parahaemolyticus* isolates from 852 fecal and environmental (water) samples. All of the 124 strains were classified into 85 known sequence types (STs), of which ST-2738 was most frequently identified. Analysis of the population structure using whole-genome variation of the 124 isolates illustrated that they grouped into 27 different clonal groups (CGs) belonging to the previously defined geographical populations VppX and VppAsia. Even though these genomes have high diversity, an extra copy of tRNA-Gly was presented in all migratory bird-carried *V. parahaemolyticus* isolates, which could be used as a potentially informative marker of the *V. parahaemolyticus* strains derived from birds. Antibiotic sensitivity experiments revealed that 47 (37.10%) isolates were resistant to ampicillin. Five isolates harbored the plasmid-mediated quinolone resistance (PMQR) gene *qnrD*, which has not previously been identified in this species. The investigation of antibiotic resistance provides the basic knowledge to further evaluate the risk of enrichment and reintroduction of pathogenic *V. parahaemolyticus* strains in migratory birds.

**IMPORTANCE** The presence of *V. parahaemolyticus* in migratory birds' fecal samples implies that the human pathogenic *V. parahaemolyticus* strains may also potentially infect birds and thus pose a risk for zoonotic infection and food safety associated with re-entry into the environment. Our study firstly highlights the extra copy of tRNA as a potentially informative marker for identifying the bird-carried *V. parahaemolyticus* strains. Also, we firstly identify the plasmid-mediated quinolone resistance (PMQR) gene *qnrD* in *V. parahaemolyticus*. To further evaluate the risk of enrichment and reintroduction of pathogenic strains carried by migratory birds, we suggest conducting estuarine environmental surveillance to monitor the antibiotic resistance and virulence factors of bird-carried *V. parahaemolyticus* isolates.

Address correspondence to Xuejun Guo, xuejung2021@163.com, Yujun Cui, cuiyujun.new@gmail.com, or Xianglilan Zhang, zhangxianglilan@gmail.com.

The authors declare no conflict of interest.

**KEYWORDS** *Vibrio parahaemolyticus*, MLST, genomic analysis, population structure, bacterial genome-wide association studies, antimicrobial resistance, virulence factors

*V*ibrio parahaemolyticus is one of the leading enteropathogenic bacteria in seafood (1). It usually causes acute and severe gastroenteritis and gastrointestinal bleeding after infection, characterized by fever, vomiting, diarrhea, and blood in the stool (2). There are four geographically related populations of *V. parahaemolyticus*, VppAisa, VppUS1, VppUS2, and VppX, identified previously, with diverse clones descending from each population (3, 4). The most common clone of *V. parahaemolyticus* among infections is serotype O3:K6 and its serovariants, leading to an epidemic worldwide since 1996 (5). With the exception of penicillin-like antibiotics, such as penicillin, ampicillin, and amoxicillin, *V. parahaemolyticus* is usually sensitive to most antibiotics of veterinary and human significance, such as chloramphenicol, tetracyclines, and quinolone (6). Although medical treatment is not necessary in mild infections, antibiotics are sometimes used in severe or prolonged illnesses (7).

Migratory birds are an important reservoir of *V. parahaemolyticus* (8). After eating food poisoned by *V. parahaemolyticus*, the birds' feces can contaminate other birds' food or even seafood (9). Additionally, during the horizontal gene transfer (HGT) process, *V. parahaemolyticus* can shape its genome by acquiring new genetic elements (10).

To further explore the *V. parahaemolyticus* spreading mechanisms from marine life to migratory birds and finally to food meant for human consumption, this study aims to characterize bird-carried *V. parahaemolyticus* strains for genetic diversity, potentially informative gene markers, and the presence of antimicrobial resistance and virulence factors. Specifically, we collected the fecal samples of migratory birds in 10 cities (11 sampling sites) of four provinces in China. We built the population structure of the newly sequenced strains in our study. We identified an extra copy of tRNA-Gly as a potentially informative marker for *V. parahaemolyticus* isolates from migratory birds, which can be used to indicate bird-carried *V. parahaemolyticus* strains. In these isolates, we also detected a plasmid with the plasmid-mediated quinolone resistance (PMQR) gene *qnrD*, which was previously found in *Proteeae*, *Salmonella enterica* subsp., *Morganella morganii*, and *Providencia stuartii* (see Table S5 in the supplemental material).

## RESULTS

**Prevalence of bird-carried *V. parahaemolyticus* in China.** We collected 852 samples from the feces of eight migratory bird species, water, and aquatic plants at 11 sampling sites (Table 1; Fig. 1). We then derived 124 *V. parahaemolyticus* strains from the *Mallards*, *Herons*, *Charadriiformes*, and water at four sampling sites in Guangdong, China, including Overseas Chinese Town Wetland Park (OCTWP) in Shenzhen, Futian Red Forest Reserve (FRFR) in Shenzhen, Leizhou, and Zhanjiang (Table 1; Fig. 1). PCR for the species-specific *tlh* gene confirmed the presence of *V. parahaemolyticus* in 124 samples, including 121 bird fecal samples and three water samples (see Table S1 in the supplemental material). Among the 124 isolates, the majority of them were recovered from the fecal samples of *Mallards* (n = 51; 41.13%) and *Charadriiformes* (n = 50; 40.32%), followed by *Herons* (n = 20; 16.13%) and water (n = 3; 2.42%). The isolation rate reached the highest (86.46%) at Shenzhen, Guangdong, China in 2019, followed by the isolation rate at Leizhou, Guangdong, China in 2019 (48.00%), the isolation rate at Shenzhen, Guangdong, China in 2018 (34.78%), and the isolation rate at Zhanjiang, Guangdong, China in 2019 (30.30%). Comparing the two sampling sites in Shenzhen, the *V. parahaemolyticus* isolation rate in OCTWP was higher than in FRFR. Specifically, in OCTWP, the isolation rate of *V. parahaemolyticus* was much higher in 2019 (97.50%) than in 2018 (37.50%); while the opposite situation was found in FRFR, where the isolation rate in 2019 (25.00%) was lower than in 2018 (28.57%).

**Genomic diversity of bird-carried *V. parahaemolyticus* in China.** The pubMLST database (11) includes 2,103 unique sequence types (STs) (March 2022). All of the 124

**TABLE 1** Prevalence of *V. parahaemolyticus* in this study

| Collection time[a] | Sample collection location[b] | Sample category | *V. parahaemolyticus* (%) | Isolation rate (%) | Isolation rate (%) based on collection time in Shenzhen |
|---|---|---|---|---|---|
| 2018.1.5 | OCTWP, Shenzhen, Guangdong, China | *Mallard* feces | 14/32 (43.75) | 37.50 | 34.78 |
| | | *Heron* feces | 4/16 (25) | | |
| | FTFR, Shenzhen, Guangdong, China | *Mallard* feces | 4/18 (22.22) | 28.57 | |
| | | *Heron* feces | 2/3 (66.66) | | |
| 2018.1.8 | Zhaoqing, Gaungdong, China | *Red-crowned crane* feces | 0/26 (0) | 0 | |
| | | *Cormorant* feces | 0/23 (0) | | |
| | | *Heron* feces | 0/18 (0) | | |
| 2018.1.13 | Beihai, Guangxi, China | *Grus japonensis* feces | 0/65 (0) | 0 | |
| | Fangchenggang, Guangxi, China | *Grus japonensis* feces | 0/91 (0) | | |
| 2018.1.14 | Nanning, Guangxi, China | *Nycticorax nycticorax* feces | 0/70 (0) | 0 | |
| 2018.1.15 | Dongxing, Guangxi, China | *Grus japonensis* feces | 0/72 (0) | 0 | |
| 2018.3.29 | Zhongning, Ningxia, China | *Mallard* feces | 0/100 (0) | 0 | |
| | | *Bone cranes* feces | 0/125 (0) | | |
| 2019.3 | OCTWP, Shenzhen, Guangdong, China | *Mallard* feces | 36/36 (100) | 97.5 | 86.46 |
| | | *Charadriiformes* feces | 40/42 (95.24) | | |
| | | Water | 2/2 (100) | | |
| | FTFR, OCT, Shenzhen, China | *Charadriiformes* feces | 4/15 (13.3) | 25.00 | |
| | | Water | 1/1(100) | | |
| 2019.3 | Zhanjiang, Guangdong, China | *Heron* feces | 8/31 (25.8) | 30.30 | |
| | | Water | 2/2 (100) | | |
| 2019.3 | Leizhou, Guangdong, China | *Heron* feces | 6/22 (27.27) | 48.00 | |
| | | *Charadriiformes* feces | 6/13 (46.15) | | |
| 2019.7 | Chifeng, Inner Mongolia autonomous region, China | *Larus ridibundus* | 0/24 | 0 | |
| | | Water | 0/4 | | |
| | | Aquatic plant | 0/1 | | |

[a]Dates are presented as year.month.day.
[b]OCTWP, Overseas Chinese Town Wetland Park; FRFR, Futian Red Forest Reserve.

strains in this study were classified into 85 STs (see Table S2 in the supplemental material). The four most frequent sequence types, ST-2738, ST-2742, ST-2693, and ST-79, possessed six, five, four, and four isolates, respectively (Table S2). There were 64 STs containing only one isolate each (Table S2).

To precisely locate the phylogenetic positions of the 124 bird-carried genomes in the *V. parahaemolyticus* species, we added 464 representative non-bird-carried *V. parahaemolyticus* strains (12) into the analysis. The 464 nonredundant *V. parahaemolyticus* strains represent our global collection of non-bird-carried *V. parahaemolyticus* strains (3) (see Table S3 in the supplemental material). In total, 588 *V. parahaemolyticus* whole genomes are considered in genomic analysis. None of the origins of non-bird-carried representative *V. parahaemolyticus* strains overlap with the sampling sites in our study. The genomes of the three water samples in our study do not show similarities with the non-bird-carried *V. parahaemolyticus* strains derived from the cities near the sampling sites (Fig. 2A, "city near/of sampling sites" strip; see also Table S3). Based on the criterion of high nucleotide identity (pairwise single nucleotide polymorphism [SNP] distance less than 2,000 SNPs) (3), the 588 whole genomes constitute 28 clonal groups (CGs), where the 124 bird-carried genomes were within 27 CGs (Fig. 2A).

We further established the population relationships across the whole data set using fineSTRUCTURE (13) based on genome-wide core SNPs (Fig. 2B). Taking the *V. parahaemolyticus* geographical populations described previously (3, 4, 12) as references, the 124 newly isolated strains were within populations VppX ($n = 15$) and VppAsia ($n = 109$). The 15 strains in population VppX were all isolated from Shenzhen (3 in FRFR and 12 in OTCWP). The sources of the 15 isolates were from *Ardeola bacchus* feces (6), *Mallard* feces (4), *Charadriiformes* feces (4), and water (1), where six samples were collected in 2018 and the other nine samples were collected in 2019. Among the 109 isolates in population VppAsia, 90 strains were isolated from Shenzhen (9 in FRFR and 81 in OTCWP), 11 strains were isolated from Leizhou, and eight strains were from Zhanjiang. The

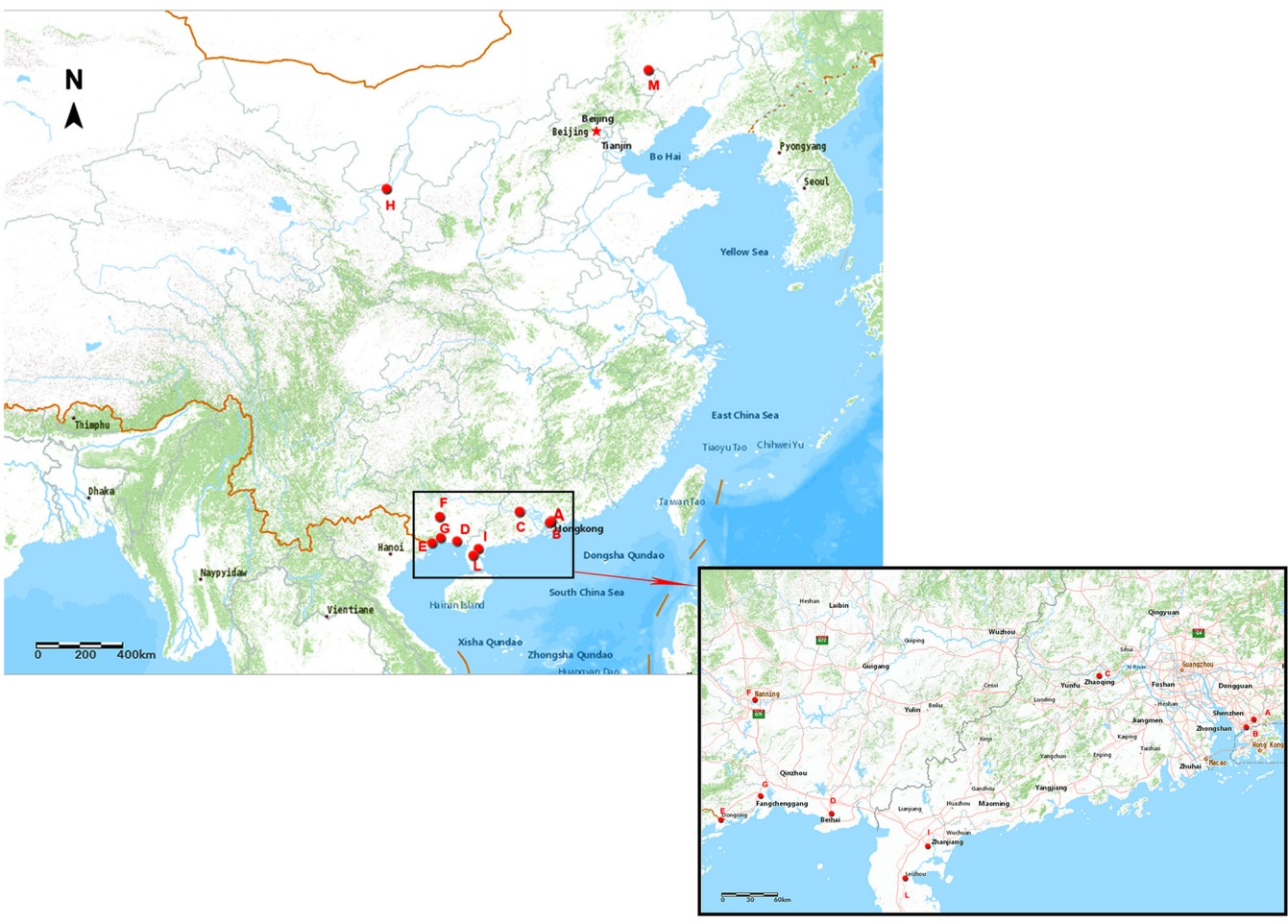

**FIG 1** This map was drawn by ArcGIS online (https://www.arcgis.com). The geographical locations of the sampling sites highlighted using red dots. (A) Overseas Chinese Town Wetland Park (OCTWP), Shenzhen, Guangdong; (B) Futian Red Forest Reserve (FRFR), Shenzhen, Guangdong; (C) Zhaoqing, Guangdong; (D) Beihai, Guangxi; (E) Fangchenggang, Guangxi; (F) Nanning, Guangxi; (G) Dongxing, Guangxi; (H) Zhongning, Ningxia; (I) Zhanjiang, Guangdong; (L) Leizhou, Guangdong; (M) Chifeng, Inner Mongolia autonomous region. All of the isolates in this study are from sites A, B, I, and L.

sources of the 109 isolates were from *Charadriiformes* feces (46), Mallard feces (43), *Ardeola bacchus* feces (18), and water (2), where 18 samples were collected in 2018 and the other 91 samples were collected in 2019.

**Genomic characteristics of bird-carried *V. parahaemolyticus* in China.** To identify the specific gene maker of bird-carried *V. parahaemolyticus* strains, we conducted a bacterial genome-wide association study (GWAS) on the sequences of 124 isolates and the 464 representative non-bird isolates representing the global distribution of *V. parahaemolyticus*. Four copies of tRNA-Gly were found in all bird-carried isolates but only in 15.09% of non-bird-carried isolates, while 84.91% of the non-bird-carried isolates contain two or three copies of tRNA-Gly (Fig. 3). The 15.09% of non-bird-carried isolates were from seven different sources in 18 different years in 11 countries of the Americas, Asia, and Europe; these isolates belong to either VppAsia or VppX (Table S3). It is noted that all bird-carried isolates contain four copies of tRNA-Gly. Therefore, we hypothesized that *V. parahaemolyticus* gene mutation occurs after the bacterial strain gets into birds' bodies. The insertion sites are the extra copies of tRNA-Gly, which are between the genes VPt086 (GenBank accession number NC_004603.1; 2,989,075 bp to 2,989,150 bp) and VPt089 (GenBank accession number NC_004603.1; 2,989,431 bp to 2,989,507 bp) in the reference genome RIMD 2210633 (Fig. 3). This kind of "four copies of tRNA-Gly" pattern is not specific for bird isolates and was also identified in non-birds isolates but with a much lower frequency (15%). As a result, the extra copy of tRNA-Gly is a potentially informative marker of bird association.

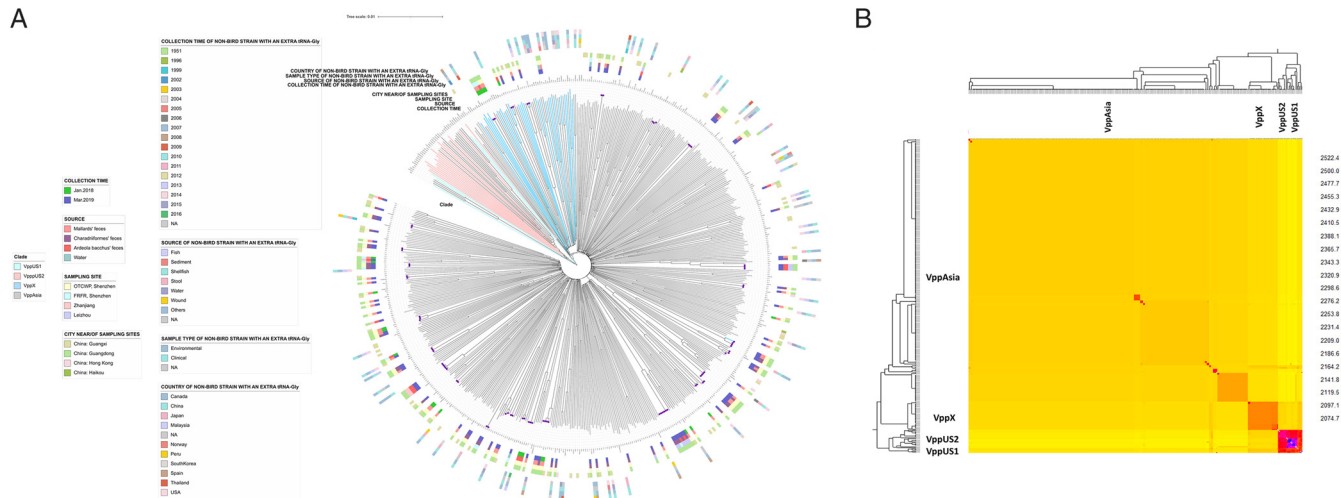

**FIG 2** Neighbor-joining tree and population structure of *V. parahaemolyticus*. (A) A core SNP-based phylogenetic tree of the 593 genomes. The colors on the clades represent population groups (VppUS1, VppUS2, VppX, and VppAsia). The purple dots at the tips of the branches show the 28 clonal groups (CGs). The location, source, and collection time of the 124 genomes and the 70 non-bird-carried *V. parahaemolyticus* genomes with an extra copy of tRNA-Gly in this study are shown by strips. (B) The population structure of the 325 genomes after grouping the strains (pairwise SNPs < 23,500). The 325 genomes represent the 588 genomes. The column represents donor strain, and the row represents recipient strain. The color of the cell shows the number of sequences that the donor transfers to the recipient. The black line indicates the lineage boundary.

**Antibiotic resistance of bird-carried *V. parahaemolyticus* in China.** To investigate antibiotic resistance of bird-carried *V. parahaemolyticus* in China, we conducted drug sensitivity analysis and antibiotic resistance gene search. All of the 124 *V. parahaemolyticus* isolates were susceptible to four beta-lactam antibiotics (meropenem, aztreonam, amoxicillin-clavulanic acid, and ampicillin/sulbactam), one quinolone (levofloxacin), tetracycline, and a sulfonamide antibiotic (trimethoprim-sulfamethoxazole). In total, six types of drug resistance genes were identified, including ampicillin resistance gene $bla_{CARB}$, quinolone resistance gene *qnr* (*qnrC* and plasmid-mediated *qnrD*), tetracycline resistance gene *tet*, fosfomycin resistance gene *fos*, trimethoprim resistance gene *dfrA*, and sulfanilamide resistance gene *sul*. The $bla_{CARB}$ gene was identified in 99.19% of isolates (123/124), and the left isolate (identification no. 34-1) encoded only *fos* (see Table S1).

Forty-six (37.10%) isolates were resistant to ampicillin, where most of them were isolated from *Charadriiformes* feces (*n* = 21; 45.65%), followed by those from *Mallard*

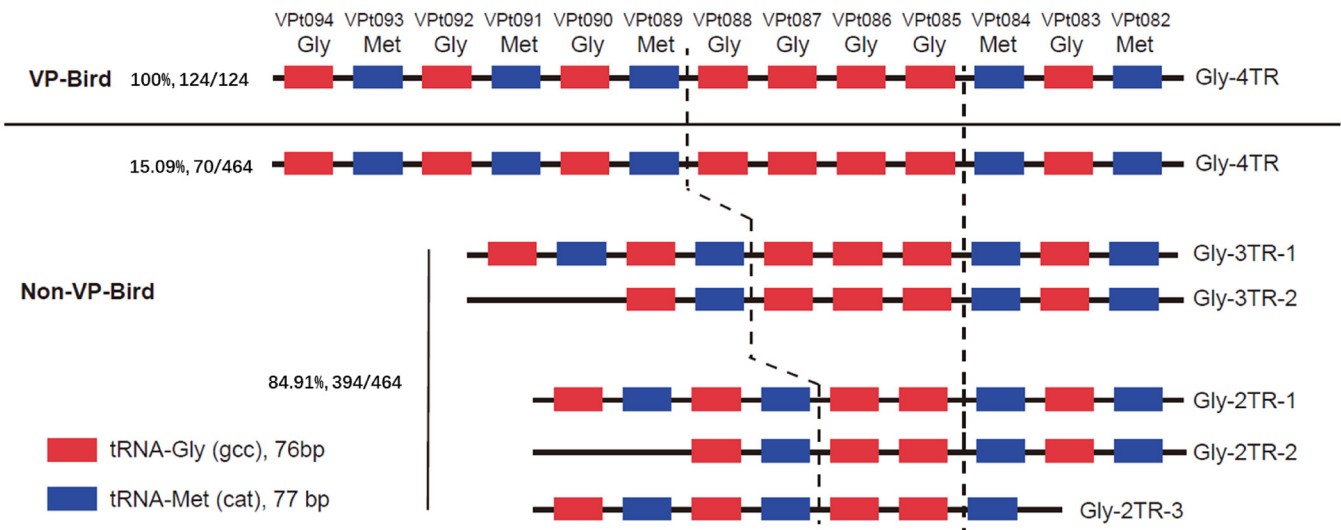

**FIG 3** The illustration of the extra copy of tRNA-Gly in bird-carried *V. parahaemolyticus* strains. VP-Bird represents the bird-carried *V. parahaemolyticus*, and Non-VP-Bird represents the non-bird-carried *V. parahaemolyticus*. VPt08x and VPt09x in the top row show the positions of the tRNA. The numbers in Gly-xTR-y in the very left column show the number of copies of tRNA-Gly (*x*) and the concatenation-style of tRNA-Gly and tRNA-Met (*y*).

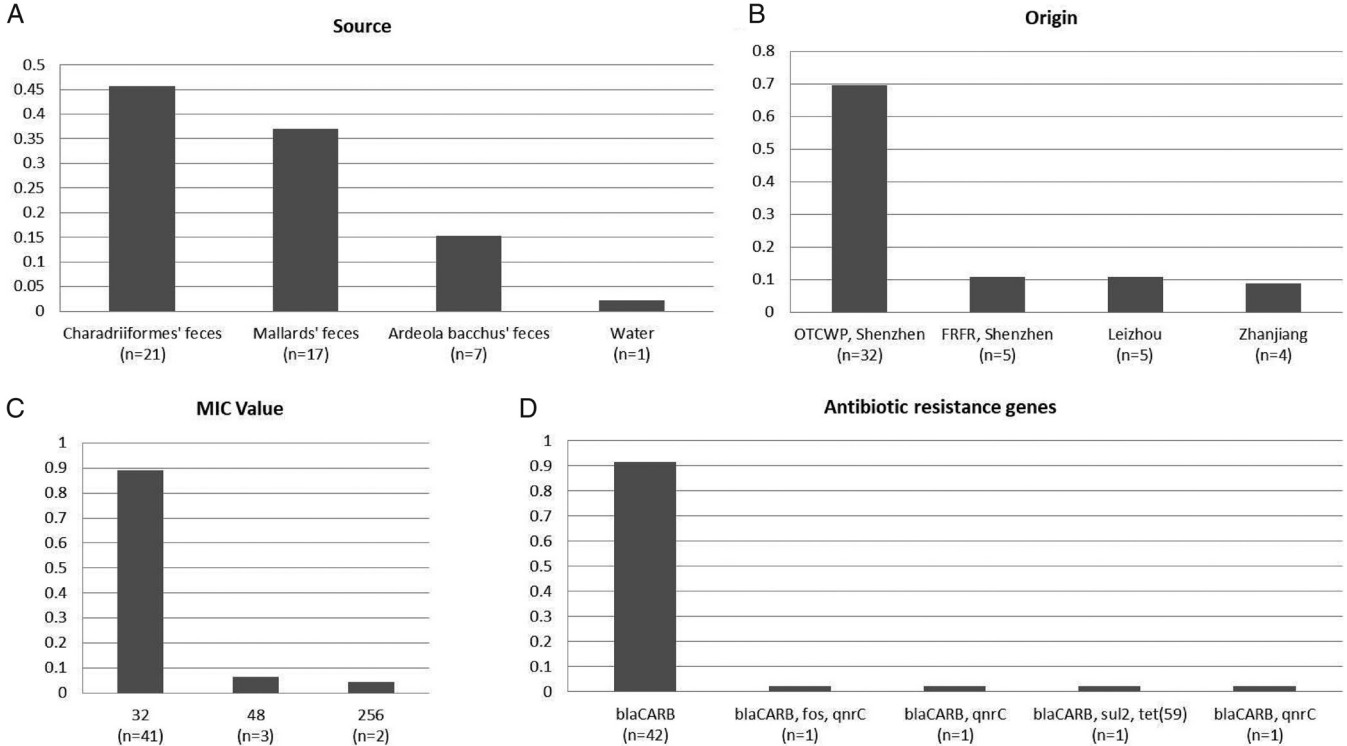

**FIG 4** Percentages of the 46 *V. parahaemolyticus* isolates resistant to ampicillin in relation to isolation source (A) and origin (B); antibiotic resistance genes (C) identified in the samples, and MIC value (D) of samples. OCTWP, Overseas Chinese Town Wetland Park; FRFR, Futian Red Forest Reserve.

feces ($n = 17$; 36.96%), *Ardeola bacchus* feces ($n = 7$; 15.22%), and water ($n = 1$; 2.17%) (Fig. 4A). Geographically, all of the 46 isolates were collected in Guangdong, China (Shenzhen [$n = 37$; 80.43%], Leizhou [$n = 5$; 10.87%], and Zhanjiang [$n = 4$; 8.70%]) (Fig. 4B). Specifically, two isolates from Shenzhen showed high resistance to ampicillin with MIC values up to 256 $\mu$g/mL (Fig. 4C). The 46 isolates all encoded ampicillin resistance genes *bla*$_{CARB}$ ($n = 46$; 100.00%), which might be related to their ampicillin resistance phenotype (Fig. 4D). By contrast, *qnrC* ($n = 3$; 6.52%), *sul2* ($n = 1$; 2.17%), *tet* ($n = 1$; 2.17%), and *fos* ($n = 1$; 2.17%) were also encoded in some of the isolates (Fig. 4D), but these isolates showed no resistance to quinolone, sulfanilamide, tetracycline, or fosfomycin (Table S1).

The genes *tdh* and *trh* were not found in any of the 124 *V. parahaemolyticus* isolates. The genes *tdh* and *trh* typically encode major virulence factors thermostable direct hemolysin (TDH) and TDH-related hemolysin (TRH), respectively. TDH is a toxin that forms pores on erythrocyte membranes, while TRH is a heat-labile toxin that is immunologically similar to TDH (14). Also, the genes *tdh* and *trh* are encoded within the pathogenicity island that harbors toxic type III secretion system T3SS2 and an array of disease-relevant secreted toxins. The absence of genes *tdh* and *trh* implied that the 124 strains are not likely human pathogens. By searching the Virulence Factor Database (VFDB) (15), none of the 124 isolates produced urease or carried a T3SS2 secretion system. They also did not harbor virulence pathogenicity islands identified in the pandemic *V. parahaemolyticus* clonal complexes O3:K6 and its serovariants (16).

Using IS Finder (17), we identified 16 types of insert components in 52 isolates collected in Shenzhen, Zhanjiang, and Leizhou of Guangdong, China (Table S1). Tn7 was the only transposon found in 12 isolates collected in Shenzhen and Zhanjiang, Guangdong, China (Table S1). None of the 124 isolates carried type I, type II, or type III integrons.

Five isolates collected in Shenzhen contained the same kind of plasmid pCol3M (Table S1; Fig. 5). The plasmid pCol3M exhibits 100.00% identity to 30 plasmids in *Proteeae*, *Providencia*, *Salmonella enterica* subsp., and *Morganella morganii* (see Table

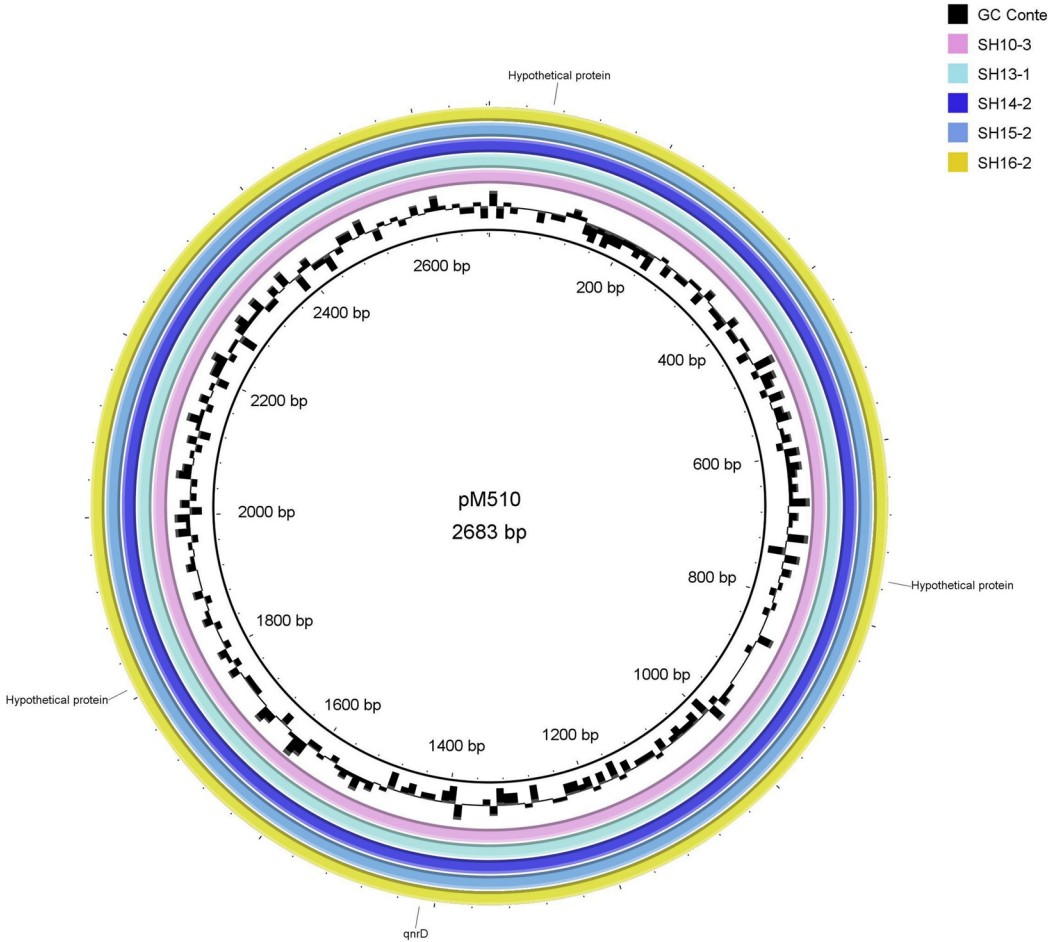

**FIG 5** Plasmid pColM within the five *V. parahaemolyticus* isolates (SH10-3, SH13-1, SH14-2, SH15-2, SH16-2) aligned to reference *P. mirabilis* plasmid pM510 (KJ190020.1). Gene locations of the hypothetical proteins and PMQR gene *qnrD* in these plasmids are indicated in the outer ring.

S5 in the supplemental material). The plasmid has 2,683 bp in length and contains three hypothetical proteins and PMQR gene *qnrD* (Fig. 5). By searching the NCBI GenBank nucleotide (nt) online database (March 2022), 65 plasmids have high similarity with pM510 (with the threshold e-value of $<1 \times 10^{-5}$ and identity/coverage of $>90\%$). The 65 plasmids belong to 20 bacterial species, including seven species in *Proteeae*, three species in *Providencia*, five species in *Salmonella enterica* subsp., etc. (Table S5).

## DISCUSSION

To further explore how the *V. parahaemolyticus* spreads between humans, animals, and the environment, this study represents comprehensive research on the characterization of *V. parahaemolyticus* isolates from the fecal samples of migratory birds toward the genetic diversity, potentially bird-carried *V. parahaemolyticus* informative gene marker, and presence of antibiotic resistance profiles.

We randomly sampled migratory birds' feces from 11 sites in Guangdong province, Guangxi Zhuang autonomous region, Ningxia Hui autonomous region, and Inner Mongolia autonomous region. In total, we recovered 124 *V. parahaemolyticus* strains from only four locations in Guangdong province, where most strains were retrieved from Shenzhen city. Interestingly, all of the 124 *V. parahaemolyticus* strains were isolated from the samples taken in Guangdong province, where most (105 strains) were collected from Shenzhen city. Unlike the Ningxia Hui autonomous region and Inner Mongolia autonomous region, Guangdong province and Guangxi Zhuang autonomous

region are located near the South China sea. The geographical difference may impact the *V. parahaemolyticus* isolation rates.

Most of the 124 *V. parahaemolyticus* strains were isolated from fecal samples of *Mallards* and *Charadriiformes*, followed by *Herons* and water. The differences in isolation rates may be due to sampling bias, which can be solved by collecting more samples in different places. Also, the bird-carried *V. parahaemolyticus* strains do not carry virulence factors in our study. These *tdh-* and *trh*-negative *V. parahaemolyticus* strains in our study were collected from southeastern China, Guangdong province, in the months of January and March (the temperature was between 18°C to 20°C). The *tdh-* and *trh*-positive *V. parahaemolyticus* strains have been reported to be isolated from *Mallard* feces (Japan in February; the temperature was below 15°C) (8) and Brown-headed gull feces (Thailand in March, September, and October; the temperature was above 20°C, according to the Thai Meteorological Department) (18). Therefore, we hypothesized that the acquisition of virulence genes (*tdh* and *trh*) may be independent of season, temperature, species of migratory birds, and geographical location.

The *V. parahaemolyticus* population has been shown to be genetically highly diverse (4, 19). Our study also revealed a high diversity of the 124 bird-carried *V. parahaemolyticus* strains belonging to 85 STs. The previous study researched the geographical populations of non-bird-carried *V. parahaemolyticus* strains, including VppX, VppAsia, VppUS1, and VppUS2 (3, 4, 12). Further genomic analysis illustrated that the 124 bird-carried strains were sparsely distributed in VppX and VppAsia, with 27 identified individual CGs, further demonstrating their high diversity on the whole-genome level. We further analyzed all of the nine publicly available genomes from bird isolates worldwide (18, 20), and six of them have the extra copy of tRNA-Gly, indicating a convergent pattern of 4× tRNA-Gly copies in bird isolates in different regions. Combined with the fact that all of the 124 bird isolates sequenced in this study also have 4× tRNA-Gly copies, the extra copy of tRNA-Gly may be useful for associating *V. parahaemolyticus* strains with birds as a potential source of contamination. As the temperate changes, *V. parahaemolyticus* varies its biological response by impacting biofilm formation (21) and gene expression (22). Therefore, the extra tRNA-Gly copy may be the biological response of *V. parahaemolyticus* to survive in birds with a higher temperature than the usual environment. Future studies could interrogate whether the extra copy of tRNA-Gly increases fitness at higher temperatures through head-to-head direct competition experiments (10, 23). It should be cautioned that *in vitro* assay is a vast simplification of the environment of a bird gut, while *in vivo* assay would help to better explore the mechanism of forming such a specifically informative gene marker.

Antibiotic resistance analysis showed that 99.19% of isolates (123/124) have $bla_{CARB}$ genes, while only 46 of the 124 isolates were tested as ampicillin resistant. These genotype-phenotype discrepancies may be due to various genetic backgrounds among unrelated isolates and epistatic interactions across genes that drive the adaption of *V. parahaemolyticus* to show different phenotypes (12, 24).

Five *V. parahaemolyticus* isolates contain plasmid pCol3M, which has a PMQR gene *qnrD*. Quinolone resistance gene *qnrD* was the latest member of the PMQR families, first described in 2009 in human clinical *Salmonella enterica* isolates in China (25). This plasmid was previously identified in *Proteeae*, *Salmonella enterica* subsp., *Morganella morganii*, and *Providencia stuartii* (see Table S5 in the supplemental material). To the best of our knowledge, this is the first study to report the plasmid with PMQR gene *qnrD* in *V. parahaemolyticus*. We suppose that this plasmid is transferable, and cross-species horizontal gene transfer events occurred to make this plasmid appear in *V. parahaemolyticus*. A possible reason for acquiring such an antibiotic resistance gene might be antibiotic overuse. Though a causal relationship between the use of conventional antibiotics and antibiotic resistance in *V. parahaemolyticus* has not been demonstrated and intrinsic resistance to some antibiotics, such as ampicillin, is possible (26, 27), antibiotic resistance could be an indication of environmental enrichment through antibiotic contamination in wastewater and runoff or use in aquaculture (22, 26, 28–31). Also, even though the five

PMQR gene-carried strains have not shown resistance to any antibiotics such as quinolone, a potential risk still exists; there could be circumstances during an infection by these strains where the PMQR gene *qnrD* confers antibiotic resistance, or this plasmid is transferred to other bacteria where it is clinically relevant.

This study indicates that *V. parahaemolyticus* may occur in migratory birds available in coastal cities of Guangdong province, Southern China, even during cold months (January and March). All of the 124 bird-carried *V. parahaemolyticus* strains were classified to 85 known STs. Further genomic analysis showed that the 124 bird-carried *V. parahaemolyticus* strains were dispersed in previously defined geographical populations (3, 4) VppX and VppAsia. Even though the 124 bird-carried *V. parahaemolyticus* strains had high diversity, an extra copy of tRNA-Gly was found in all of them, an indicator for bird-borne environmental and food contamination. This study firstly reports *V. parahaemolyticus* isolates being positive for the PMQR gene, quinolone resistance gene *qnrD*, which was embedded in the plasmid pCol3M.

Also, our study supplements the worldwide antibiotic resistance profiles of *V. parahaemolyticus* isolated from migratory birds. Previous studies have shown that the migratory birds traveling to Bangladesh and Romania potentially carry multidrug-resistant *Vibrio* spp. (32, 33). Our study illustrated the antibiotic resistance profile of *V. parahaemolyticus* strains isolated from migratory birds in Guangdong, China, which were resistant to ampicillin in both genotype and phenotype.

## MATERIALS AND METHODS

**Sampling.** From 5 January to 29 March 2018 and from 13 March to 13 July 2019, 842 samples of feces, seven water samples, and one aquatic plant sample were collected from Guangdong province (Shenzhen, Zhaoqing, Zhanjiang, Leizhou), Guangxi Zhuang autonomous region (Beihai, Fangchengxiang, Nanning, Dongxing), Ningxia Hui autonomous region (Zhongning), and Inner Mongolia autonomous region (Chifeng) (Table 1). In addition, we also determined a range of environmental variables (water temperature, pH, and salinity). DNA was obtained from each fecal sample by stool DNA kit. The host sources of the feces were determined by the molecular method described in reference 34.

***V. parahaemolyticus* isolation and identification.** The isolation and identification of *V. parahaemolyticus* were conducted based on the methods described in the National Food Safety Standards of China (GB 4789.7-2013). Briefly, 25-g feces samples from each bird were homogenized with 225 mL of phosphate-buffered saline (PBS) for 15 to 30 s in a 4-mL centrifuge tube, followed by incubation at 37°C for 16 h. One hundred microliters mixed liquid was plated on CHROM agar *Vibrio* plates (CHROM, Paris, France) and cultivated at 37°C for 16 h. One purple isolate (round, translucent, smooth surface, diameter of 2 to 3 mm) was randomly selected as a *V. parahaemolyticus* candidate and thus picked for PCR identification from each CHROM agar *Vibrio* plate. Specifically, the purple colonies of each strain were first cultured overnight. Then, 2 mL overnight grown culture was collected by centrifugation, the cell pellets were resuspended in 200 $\mu$L of double-distilled water (ddH$_2$O) buffer, and heated in the water bath for 7 min. Finally, the supernatant (DNA template) was collected and stored at $-20$°C for PCR, where the cycling number was set as 30 and the annealing temperatures were set as 55°C (*toxR*), 60°C (*tdh*, *trh*, and *tlh*), and 52°C (Intl1 and 2), separately (see Table S4 in the supplemental material).

A total of 129 (129/842; 14.55%) samples, including 124 migratory bird fecal samples and five water samples, tested positive for *V. parahaemolyticus* due to biochemical identification (Table 1). The PCR for the species-specific *tlh* gene confirmed the presence of these bacteria in 124 (124/129; 96.12%) samples, including 121 birds' fecal samples and three water samples (see Table S1 in the supplemental material). Finally, the 124 strains encoding the *tlh* gene were identified as *V. parahaemolyticus* isolates and analyzed in our study. The accession numbers of these 124 *V. parahaemolyticus* genome assemblies are listed in Table S1.

**MIC determination.** The sensitivity analysis included 18 antibiotics (2 aminoglycosides, 10 beta-lactams, 1 chloramphenicol, 3 quinolones, 1 tetracycline, 1 sulfonamide). The MIC value was determined by the Etest strips method. All strains needed Mueller-Hinton (MH) liquid to adjust their concentration to 0.5 McFarland turbidity; the reference strain was ATCC 25922. All adjusted strains were tested in a flat containing 25-mm-thick MH solid medium, and then mediums were incubated at 37°C for 16 h. The final interpretation and the susceptibility results were confirmed according to the guideline (35).

**Extraction of genome DNA and genome sequencing.** Genomic DNA was extracted with a bacterial genomic DNA extraction kit (Omega). The harvested DNA was detected by a Qubit 2.0 fluorometer (Thermo Scientific). Sequencing libraries were generated using NEBNext Ultra DNA library prep kit for Illumina (NEB, USA) following the manufacturer's recommendations, and index codes were added to attribute sequences to each sample. Briefly, the DNA sample was fragmented by sonication to a size of 350 bp. DNA fragments were then end-polished, a-tailed, and ligated with the full-length adaptor for Illumina sequencing with further PCR amplification. At last, PCR products were purified (AMPure XP system), and libraries were analyzed for size distribution by an Agilent 2100 Bioanalyzer and quantified

using real-time PCR. The genome of *V. parahaemolyticus* was sequenced using Illumina NovaSeq PE150 at the Beijing Novogene Bioinformatics Technology Co., Ltd.

**Genome assembly.** Trimmomatic V10 (36) was used to remove the PCR adapters and low-quality reads. SPAdes (37) (http://cab.spbu.ru/software/spades/) was used to do sequence assembly, and the assembled genomes were annotated using Prokka (38). Both assembly-based and assembly-free algorithms were used to improve SNP calling accuracy. Specifically, we identified SNPs by integrating assembly-based method MUMmer version 3.23 (39) and assembly-free method Snippy (40). The SNPs identified by both methods were considered as high-quality SNPs and were used for further phylogenetic analysis. As a result, reads with average Phred quality score less than 20 were removed, and the average Phred quality scores of filtered reads were higher than 30. $N_{50}$, number of contigs (NrContigs), and length of the assembled genome are as follows: the average $N_{50}$ is 460,970, and the average number of contigs and the average size of assemblies were 90 and 5,123,946 bp, respectively. The average number of congruent alleles (NrConsensus) is 2,254. A total of 690,261 high-quality SNPs were identified, with the CorePercent of 97.4%. Detail information is provided in Table S6 in the supplemental material.

**Plasmid comparison.** The contigs representing plasmids in the five *V. parahaemolyticus* bacterial strains were pooled and submitted to the BLAST Ring Image Generator (BRIG) (41) tool for mapping. Using BRIG, contigs were aligned to *Proteus mirabilis* plasmid pM510 (KJ190020.1) using the blastn search option and a sequence identity cutoff of ≥95%.

**Gene annotation and mobile element identification.** Antimicrobial resistance genes in the genome of *V. parahaemolyticus* were predicted by CARD (42) (https://card.mcmaster.ca/about). Virulence factors in the genome of *V. parahaemolyticus* were identified using the VFDB (15) (http://www.mgc.ac.cn./VFs/main.htm). PlasmidFinder (43) (https://cge.food.dtu.dk/services/PlasmidFinder/) was used to predict the presence of the plasmid. The ISfinder (17) (http://issaga.biotoul.fr/ISsaga2/issaga_index.php) was used to identify insert elements.

**Multilocus sequence typing.** The sequences of seven conserved housekeeping genes, including *dnaE*, *gyrB*, *recA*, *dtds*, *pntA*, *pyrC*, and *tnaA*, and the ST of each *V. parahaemolyticus* strain were obtained by uploading its assembly to the pubMLST. Particularly, on pubMLST (https://pubmlst.org/), we first typed "*Vibrio parahaemolyticus*" in the search bar and then clicked "Typing" to enter the webpage of "*Vibrio parahaemolyticus* typing database," where we submitted the *V. parahaemolyticus* assemblies with default parameters ("All loci" for "locus/scheme" and "order results by" "locus").

**Genome data set, SNP calling, and genomic analysis of population structure.** Our previous research has chosen 464 representative non-bird-carried *V. parahaemolyticus* whole genomes, constituting geographical populations VppUS1, VppUS2, VppX, and VppAsia (3, 4). To precisely classify the newly found 124 bird-carried *V. parahaemolyticus* strains to predefined geographical populations, we included the 464 representative strains in our study. The 464 nonredundant *V. parahaemolyticus* strains were generated by our global collection of 1,103 isolates (3), representing the global distribution of non-bird-carried *V. parahaemolyticus*. SNPs were identified as previously described (4, 12, 24). Briefly, the 588 assembled sequences (124 strains in this study and 464 representative strains) were aligned against a reference genome RIMD 2210633 (NC_004603.1 and NC_004605.1) using MUMmer (44) to generate the whole-genome alignments and identify SNPs in the core genome (regions presented in all isolates). From the 588 isolates, we identified 708,470 SNPs that were used in the neighbor-joining tree construction using TreeBeST 1.9.2 (http://treesoft.sourceforge.net/treebest.shtml). The phylogenetic tree was visualized using iToL (45).

By calculating the pairwise SNP distance between all of the 588 isolates, we defined 28 CGs among the sequences whose genome differences are less than 2,000 SNPs (3), with each CG including two to six strains. The 124 newly isolated strains in this study were within 27 CGs. To define the population structure of the isolates while excluding the effect of clonality, we used an iterative algorithm to successively remove strains with clonal relatedness (3). Finally, a total of 325 genomes (pairwise SNPs > 23,500) were left to run Chromosome painting and fineSTRUCTURE (13).

**Genome-wide association analysis.** Pyseer (46) can capture SNPs, indels, and accessory genome elements. We used pyseer for k-mer-based genome-wide association analysis to identify bird-associated gene makers with default settings. Previously published 464 representative human and food isolates were included as non-bird isolates (3, 4). Bird and non-bird isolates were treated as binary phenotypes. A $P$ value of 0.05 was set as the significance threshold after the Bonferroni correction. Significant k-mers with the lowest $P$ values were mapped to a tRNA region (VPt085-VPt088, tRNA-Gly) in the reference genome RIMD 2210633 (NC_004603.1, NC_004605.1).

**Data availability.** All data generated or analyzed during this study are available from the corresponding author on reasonable request.

The genome sequences were archived to the sequence repository GenBank and their accession numbers are listed in Table S1 in the supplemental material. All of the 124 *V. parahaemolyticus* genome assemblies are available at BioProject accession number PRJNA786264.

## SUPPLEMENTAL MATERIAL

Supplemental material is available online only.
**SUPPLEMENTAL FILE 1**, PDF file, 0.27MB.

## ACKNOWLEDGMENTS

This study was supported by the National Key R&D Program of China (no. 20212302004) and the Natural Science Foundation of China (no. 31770001).

We are grateful to members of the Wild Animal Sources and Diseases Inspection Station, National Forestry, and Grassland Bureau.

Y. Cui, X. Guo, P. Chen, and L. Zhu conceived and supervised the project. L. Zhu, S. Sun, Y. Wang, and G. Lu collected the samples. M. Liu, J. Jie, and L. Zheng conducted the drug sensitivity analysis. X. Zhang, C. Yang, Y. Cui, J. Guan, and H. Wen performed bioinformatics analysis. X. Zhang and L. Zheng wrote the manuscript. R. Yang and Y. Song provided insightful comments and revised the manuscript. All authors approved the final manuscript.

We declare that we have no competing interests.

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
