## [Reviewer comments · Microbiology Spectrum]

Microbiology Spectrum

***Vibrio parahaemolyticus* from Migratory birds in China carries an extra copy of tRNA-Gly and plasmid-mediated quinolone resistance (PMQR) gene qnrD**

Lin Zheng, Chao Yang, Ping Chen, Lingwei Zhu, Huiqi Wen, Mingwei Liu, Jia-yao Guan, Gejin Lu, Jing Jie, Shiwen Sun, Ying Wang, Yajun Song, Ruifu Yang, Xianglilan Zhang, Yujun Cui, and XueJun Guo

Corresponding Author(s): XueJun Guo, Changchun Veterinary Research Institute, Chinese Academy of Agricultural Sciences/Key laboratory of Jilin Province for Zoonosis Prevention and Control

Review Timeline:

Submission Date:	June 14, 2022
Editorial Decision:	December 20, 2022
Revision Received:	February 14, 2023
Accepted:	May 5, 2023

Editor: Gaurav Sharma

Reviewer(s): Disclosure of reviewer identity is with reference to reviewer comments included in decision letter(s). The following individuals involved in review of your submission have agreed to reveal their identity: Clifford G. Clark (Reviewer #2)

Transaction Report:

DOI: <https://doi.org/10.1128/spectrum.02170-22>

December 20, 2022

Dr. Xuejun Guo
Changchun Veterinary Research Institute, Chinese Academy of Agricultural Sciences/Key laboratory of Jilin Province for Zoonosis Prevention and Control
Changchun
China

Spectrum02170-22 (*Vibrio parahaemolyticus* from Migratory birds in China carries an extra copy of tRNA-Gly and plasmid-mediated quinolone resistance (PMQR) gene qnrD)

Dear Dr. Xuejun Guo,

Thank you for submitting your manuscript to Microbiology Spectrum. Reviewers have taken a lot of time in sending their reviews and considering that, I want to specially thank you for your patience. Now we have received the comments from two reviewers for your submitted manuscript. Both reviewers have pointed out major concerns and recommended additional experiments and explanations. Therefore, I invite you to revise your manuscript in light of the referees' comments. Your manuscript needs major revisions and the comments from the referees are appended below for your attention. Please ensure that the added explanations and comments in the rebuttal letter are well explained throughout the text and supported by clearly described methods.

Link Not Available

Sincerely,

Gaurav Sharma, Ph.D.
Assistant Professor, IIT Hyderabad
Personal Lab Page: <https://sites.google.com/view/sharmaglab/>
Editor, Microbiology Spectrum

Journals Department
Reviewer comments:

Reviewer #2 (Comments for the Author):

In this interesting manuscript the authors have tested birds for carriage of *Vibrio parahaemolyticus*. Isolates were recovered from birds, sequenced, and compared to isolates from sources other than birds. The results suggest that birds may be a vector for the

spread of the bacterium between aquatic environments and other sources, including fish, though they also suggest that the bird *V. parahaemolyticus* may constitute a bird-adapted variant.

You have stated the purpose of the research as characterization of the isolates infecting birds and searching for genetic markers for bird-specific isolates. Was there a more general hypothesis or a larger question to answer behind the decision to undertake this research? If so, explicitly stating it would help to provide the larger context for the research.

Two methodological processes would benefit greatly from additional detail. Most people that routinely work with archival sequence data know that it is necessary to apply quality parameters to sort out bad sequence from good.

1) I think it is a good idea to provide genome sequence quality data in the manuscript, at minimum the average sequence quality (AvgQuality), N50, number of contigs (NrContigs), length of assembled genome, the congruent alleles by both assembly-based and assembly-free algorithms (NrConsensus), and the CorePercent. These should be available from the software generating the assemblies. It should be explicitly stated if assembly-free algorithms were not used.

2) When doing SNP analysis for creating phylogenetic trees it is best practice to use only high quality SNPs. Please include a brief explanation how high quality SNPs were identified and used.

Genome sequences should be archived in an appropriate sequence repository and accession numbers provided in the manuscript.

There is an instance in lines 160-162 where you state that the data you provide "implies" that a specific process is occurring. "It is noted that all bird-carried isolates contain four copies of tRNA-Gly, implying that *V. parahaemolyticus* gene mutation occurs after the bacterial strain gets into birds' bodies." As there is currently no data to support this, it would definitely be better to state this as an hypotheses. At a later point you do suggest ways to test the hypothesis, which is very good.

I would be interested in seeing the insertion sites of the different tRNA-Gly instances. Is the additional site "new" and not seen in isolates from sources other than birds Or are there already four (or more) insertion sites and only two to three are occupied in isolates from sources other than birds while four are occupied only in isolates from birds?

Reviewer #3 (Comments for the Author):

Authors have shown the genetic feature of *V. parahaemolyticus* isolated from birds with large number of isolate collection (n=124). The analysis of bird origin by PCR marker of bird species was comprehensive. Geographical distribution of bird isolates source was well discussed. Main findings included the presence of extra copy tRNA-Gly and qnrD gene in bird isolates. However, there are few questions/issues are concerned as following:

1. In material and method (line 309), "The 25g fecal samples were homogenized..." Was the 25g of sample obtained from pooled fecal samples of different birds or from one single bird? Please clarify.

2. In material and method (line 313-314), how many isolate was selected from each fecal sample? Assumably, it was one isolate from one sample (?). Then, what was the criteria of choosing an isolate among other purple *Vp* colonies in the CHROMagar plate?

3. In material and method (line 328), the accession numbers of genome assemblies of 124 isolates are absent from Table S1. Please double check.

4. Author addressed that the extra copy of tRNA-Gly is restricted only in bird isolates. It would be more convincible if the bird isolates from other part of the world are included. A number of previous papers have reported detection of *V. parahaemolyticus* in aquatic birds with the publicly available genome sequences (e.g. Fu, 2019, DOI: 10.1038/s41598-019-52791-5; Muangnapoh, 2022, DOI:10.1128/spectrum.00886-22).

Without including other worldwide bird isolates, the statement "...implying that *Vp* gene mutation occurs after the bacterial strain gets into bird's body..." (line 160-163) is quite skeptical. In fact, the tRNA-Gly was also detected in non-bird isolates (15.09%) in this study although it was much smaller number than those in the bird isolates. In this way, the worldwide bird *Vp* are indeed necessary to be incorporated into the analysis to be strongly suggest that the extra copy of tRNA-Gly is potentially a gene marker for bird *Vp*.

5. The tdh+ and/or trh+ *Vp* were not detected from this study. However, previous studies have reported the tdh+ and/or trh+ *Vp* from bird although the number of isolates were smaller compared to the isolates in this study (Miyasaka, 2006, DOI: 10.1017/S0950268805005674; Muangnapoh, 2022, DOI:10.1128/spectrum.00886-22). The discussion for this part should be added for more understanding about the situation facilitating transmission of pathogenic *Vp* by birds. For example, the seasonal factor probably affect in the absence of pathogenic *Vp* in this study as the sample were collected during the cold month(?).

Normally, the pathogenic Vp favor warmer growth temperature compared to the non-pathogenic Vp.

6. The template of the references list is inconsistency (e.g. use of capital/small letter, journal abbreviation, italics where needed). It should be revised carefully.

Staff Comments:

Preparing Revision Guidelines

Please return the manuscript within 60 days; if you cannot complete the modification within this time period, please contact me. If you do not wish to modify the manuscript and prefer to submit it to another journal, please notify me of your decision immediately so that the manuscript may be formally withdrawn from consideration by Microbiology Spectrum.

Dear Editor,

Thank you for giving us the opportunity to submit our revised manuscript "*Vibrio parahaemolyticus* from Migratory birds in China carries an extra copy of tRNA-Gly and plasmid-mediated quinolone resistance (PMQR) gene *qnrD*". We appreciate the time and effort that you and the reviewers dedicated to providing feedback on our manuscript and are very grateful for the insightful comments on and valuable improvements to our paper. We have incorporated all the suggestions highlighted as tracked changes within our manuscript. Also, please see below, we have made a point-by-point response to each of the reviewers' comments.

Reviewer comments:

Reviewer #2 (Comments for the Author):

In this interesting manuscript the authors have tested birds for carriage of *Vibrio parahaemolyticus*. Isolates were recovered from birds, sequenced, and compared to isolates from sources other than birds. The results suggest that birds may be a vector for the spread of the bacterium between aquatic environments and other sources, including fish, though they also suggest that the bird *V. parahaemolyticus* may constitute a bird-adapted variant.

Response: Thank Reviewer #2 for your time and effort in providing such valuable suggestions. We have updated our manuscript to address all of the comments and concerns. Please see below, the point-by-point response to these comments.

You have stated the purpose of the research as characterization of the isolates infecting birds and searching for genetic markers for bird-specific isolates. Was there a more general hypothesis or a larger question to answer behind the decision to undertake this research? If so, explicitly stating it would help to provide the larger context for the research.

Response: We have revised the abstract and introduction in our manuscript:

Abstract (lines 36-40): "To further explore the spreading mechanism of *V. parahaemolyticus* among marine, human beings, and migratory birds, we aimed to investigate the characteristics of the genetic diversity, antimicrobial resistance, virulence genes, and potentially informative gene marker of *V. parahaemolyticus* isolated from migratory birds in China."

Importance (lines 56-59): "The presence of *V. parahaemolyticus* in migratory birds' fecal samples implies that the human pathogenic *V. parahaemolyticus* strains may also potentially infect birds and thus pose a risk for zoonotic infection and food safety associated with re-entry into the environments."

Introduction (lines 87-92): "To further explore the *V. parahaemolyticus* spreading mechanisms from marine, to the migratory birds, and finally to the food table, this study aims to characterize bird-carried *V. parahaemolyticus* strains for genetic diversity, potentially informative gene marker, and the presence of antimicrobial resistance and virulence factors. Specifically, we collected the fecal samples of migratory birds in ten cities (11 sampling sites) of four provinces in China."

Discussion (lines 226-230): "To further explore how the *V. parahaemolyticus* spreads between humans, animals, and the environment, this study represents comprehensive research on the characterization of *V. parahaemolyticus* isolates from the fecal samples of migratory birds towards the genetic diversity, potentially bird-carried *V. parahaemolyticus* informative gene marker, and presence of antibiotic-resistant profiles."

Two methodological processes would benefit greatly from additional detail. Most people that routinely work with archival sequence data know that it is necessary to apply quality parameters to sort out bad sequence from good.

1) I think it is a good idea to provide genome sequence quality data in the manuscript, at minimum the average sequence quality (AvgQuality), N50, number of contigs (NrContigs), length of assembled genome, the congruent alleles by both assembly-based and assembly-free algorithms (NrConsensus), and the CorePercent. These should be available from the software generating the assemblies. It should be explicitly stated if assembly-free algorithms were not used.

Response: We have added the data quality information in the method and results sections of our manuscript:

Materials and methods (lines 376-386): "Both assembly-based and assembly-free algorithms were used to improve SNP calling accuracy. Specifically, we identified SNPs by integrating assembly-based method MUMmer version 3.23 (38) and assembly-free method Snippy (39). The SNPs identified by both methods were considered as high-quality SNPs and used for further phylogenetic analysis. As a result, reads with average Phred quality score less than 20 were removed and the average Phred quality scores of filtered reads were higher than 30. N50, number of contigs (NrContigs), length of the assembled genome: the average N50 is 460,970, the average number of contigs and the average size of assemblies were 90 and 5,123,946 bp, respectively. The average number of congruent alleles (NrConsensus) is 2,254. A total of 690,261 high-quality SNPs were identified, with the CorePercent of 97.4%. The detail information is provided in Table S6. "

References:

36. Bolger, A.M., Lohse, M. and Usadel, B. (2014) Trimmomatic: a flexible trimmer for Illumina sequence data. *Bioinformatics*, **30**, 2114-2120.

39. Kurtz, S., Phillippy, A., Delcher, A.L., Smoot, M., Shumway, M., Antonescu, C. and Salzberg, S.L. (2004) Versatile and open software for comparing large genomes. *Genome Biol.*, **5**, 1-9.

40. Seemann, T. Snippy. <https://github.com/tseemann/snippy>.

2) When doing SNP analysis for creating phylogenetic trees it is best practice to use only high quality SNPs. Please include a brief explanation how high quality SNPs were identified and used.

Response: We have further explained the SNP choice in the materials and methods section of our manuscript (lines 376-380):

"Both assembly-based and assembly-free algorithms were used to improve SNP calling accuracy. Specifically, we identified SNPs by integrating assembly-based method MUMmer version 3.23

(37) and assembly-free method Snippy (38). The SNPs identified by both methods were considered as high-quality SNPs and used for further phylogenetic analysis."

References

39. Kurtz, S., Phillippy, A., Delcher, A.L., Smoot, M., Shumway, M., Antonescu, C. and Salzberg, S.L. (2004) Versatile and open software for comparing large genomes. *Genome Biol.*, **5**, 1-9.
40. Seemann, T. Snippy. <https://github.com/tseemann/snippy>.

Genome sequences should be archived in an appropriate sequence repository and accession numbers provided in the manuscript.

Response: We have added the accession numbers of the genome sequences in our manuscript (lines 447-449):

"The genome sequences were archived to the sequence repository GenBank (<http://www.ncbi.nlm.nih.gov/genbank/>) and their accession numbers are listed in TABLE S1."

There is an instance in lines 160-162 where you state that the data you provide "implies" that a specific process is occurring. "It is noted that all bird-carried isolates contain four copies of tRNA-Gly, implying that *V. parahaemolyticus* gene mutation occurs after the bacterial strain gets into birds' bodies." As there is currently no data to support this, it would definitely be better to state this as an hypothesis. At a later point you do suggest ways to test the hypothesis, which is very good.

Response: We have revised this description to a hypothesis in our manuscript (lines 164-167):

"It is noted that all bird-carried isolates contain four copies of tRNA-Gly. Therefore, we hypothesized that *V. parahaemolyticus* gene mutation occurs after the bacterial strain gets into birds' bodies."

I would be interested in seeing the insertion sites of the different tRNA-Gly instances. Is the additional site "new" and not seen in isolates from sources other than birds Or are there already four (or more) insertion sites and only two to three are occupied in isolates from sources other than birds while four are occupied only in isolates from birds?

Response: We have added the description of insertion sites in our manuscript (lines 167-172):

"The insertion sites are the extra copies of tRNA-Gly, which are between the genes VPt086 (NC_004603.1: 2,989,075 bp - 2,989,150 bp) and VPt089 (NC_004603.1: 2,989,431 bp - 2,989,507 bp) in the reference genome RIMD 2210633 (FIG 3). This kind of "four copies of tRNA-Gly" pattern is not specific for bird isolates, which was also identified in non-birds isolates but with a much lower frequency (15%)."

Reviewer #3 (Comments for the Author):

Authors have shown the genetic feature of *V. parahaemolyticus* isolated from birds with large

number of isolate collection (n=124). The analysis of bird origin by PCR marker of bird species was comprehensive. Geographical distribution of bird isolates source was well discussed. Main findings included the presence of extra copy tRNA-Gly and *qnrD* gene in bird isolates. However, there are few questions/issues are concerned as following:

Response: Thank Reviewer #3's valuable suggestions. We have revised our manuscript according to the advice. Please see the following responses to each of the comments.

1. In material and method (line 309), "The 25g fecal samples were homogenized..." Was the 25g of sample obtained from pooled fecal samples of different birds or from one single bird? Please clarify.

Response: Revised (lines 329-331):

"Briefly, 25g fecal samples from each bird were homogenized with 225mL of phosphate-buffered saline (PBS) for 15–30s in a 4 mL centrifuge tube, followed by incubation at 37°C for 16 h."

2. In material and method (line 313-314), how many isolate was selected from each fecal sample? Assumably, it was one isolate from one sample (?). Then, what was the criteria of choosing an isolate among other purple *Vp* colonies in the CHROMagar plate?

Response: Revised:

According to the pre-experimental results and operation instruction of CHROM agar *Vibrio* plate, above 90% of the round, semi-transparent, smooth surface and 2-3mm diameter isolates on the CHROM agar *Vibrio* plate were *V. parahaemolyticus* colonies, so we revised as follow:

Lines 333-335: "One purple isolate (round, translucent, smooth surface, diameter 2-3 mm) was randomly selected as *V. parahaemolyticus* candidates and thus picked for PCR identification from each CHROM agar *Vibrio* plate."

3. In material and method (line 328), the accession numbers of genome assemblies of 124 isolates are absent from Table S1. Please double check.

Response: We have added the accession numbers of the 124 genome assemblies in our manuscript (lines 447-449):

"The genome sequences were archived to the sequence repository GenBank (<https://www.ncbi.nlm.nih.gov/genbank/>) and their accession numbers are listed in TABLE S1."

4. Author addressed that the extra copy of tRNA-Gly is restricted only in bird isolates. It would be more convincible if the bird isolates from other part of the world are included. A number of previous papers have reported detection of *V. parahaemolyticus* in aquatic birds with the publicly available genome sequences (e.g. Fu, 2019, DOI: 10.1038/s41598-019-52791-5; Muangnapoh, 2022, DOI:10.1128/spectrum.00886-22).

Without including other worldwide bird isolates, the statement "...implying that *Vp* gene mutation occurs after the bacterial strain gets into bird's body..." (line 160-163) is quite skeptical. In fact, the tRNA-Gly was also detected in non-bird isolates (15.09%) in this study although it was much

smaller number than those in the bird isolates. In this way, the worldwide bird *Vp* are indeed necessary to be incorporated into the analysis to strongly suggest that the extra copy of tRNA-Gly is potentially a gene marker for bird *Vp*.

Response: We have analyzed the worldwide bird *V. parahaemolyticus* strains and added the discussion in our manuscript (lines 262-276):

"We further analyzed all the nine publicly available genomes from bird isolates worldwide (18,20), and six of them have the extra copy of tRNA-Gly, indicating a convergent pattern of 4 x tRNA-Gly copies in bird isolates in different regions. Combined with the fact that all the 124 bird isolates sequenced in this study also have 4 x tRNA-Gly copies, the extra copy of tRNA-Gly may be useful for associating *V. parahaemolyticus* strains with birds as a potential source of contamination. As the temperate changes, *V. parahaemolyticus* varies its biological response by impacting biofilm formation (21) and gene expression (22). Therefore, the extra tRNA-Gly copy may be the biological response of *V. parahaemolyticus*, to survive in birds with a higher temperature than the usual environment. Future studies could interrogate whether the extra copy of tRNA-Gly increases fitness at higher temperatures through head-to-head direct competition experiments (10,23). It should be cautious that *in vitro* assay is a vast simplification of the environment of a bird gut; while *in vivo* assay would help better explore the mechanism of forming such a specifically informative gene marker."

Public genomes of strains from birds.

Geographical region	Strain ID	Accession number	Copy number of tRNA-Gly	Reference
Panjin, China	PJ18	SRR8892844	4	Fu, 2019*
	PJ35	SRR8892846	2	
	PJ37	SRR8892845	2	
Yinkou, China	YK33	SRR8892853	2	
Thailand	MAVP8	JALAZC000000000	4	Muangnapoh, 2022
	MAVP9	JALAZB000000000	4	
	MAVP10	JALAZA000000000	4	
	MAVP20	JALAYZ000000000	4	
	MAVP22	JALAYY000000000	4	

*In Fu.2019, a total of seven strains from birds were sequenced, with only four of them being publicly available.

References

- Nyholm, S.V. and McFall-Ngai, M.J. (2003) Dominance of *Vibrio fischeri* in secreted mucus outside the light organ of *Euprymna scolopes*: the first site of symbiont specificity. *Appl. Environ. Microbiol.*, **69**, 3932-3937.
- Muangnapoh, C., Tamboon, E., Supha, N., Toyting, J., Chitrak, A., Kitkumthorn, N., Ekcharyawat, P., Iida, T. and Suthienkul, O. (2022) Multilocus sequence typing and virulence potential of *Vibrio parahaemolyticus* strains isolated from aquatic bird feces. *Microbiol. Spectr.*, **10**, e00886-00822.
- Fu, S., Hao, J., Yang, Q., Lan, R., Wang, Y., Ye, S., Liu, Y. and Li, R. (2019) Long-distance transmission of pathogenic *Vibrio* species by migratory waterbirds: a potential threat to the public

health. *Sci. Rep.*, **9**, 16303.

21. Song, X., Ma, Y., Fu, J., Zhao, A., Guo, Z., Malakar, P.K., Pan, Y. and Zhao, Y. (2017) Effect of temperature on pathogenic and non-pathogenic *Vibrio parahaemolyticus* biofilm formation. *Food Control*, **73**, 485-491.
22. Urmersbach, S., Aho, T., Alter, T., Hassan, S.S., Autio, R. and Huehn, S. (2015) Changes in global gene expression of *Vibrio parahaemolyticus* induced by cold-and heat-stress. *BMC Microbiol.*, **15**, 1-13.
23. Kalburge, S.S., Carpenter, M.R., Rozovsky, S. and Boyd, E.F. (2017) Quorum sensing regulators are required for metabolic fitness in *Vibrio parahaemolyticus*. *Infect. Immun.*, **85**, e00930-00916.

5. The *tdh+* and/or *trh+* *Vp* were not detected from this study. However, previous studies have reported the *tdh+* and /or *trh+* *Vp* from bird although the number of isolates were smaller compared to the isolates in this study (Miyasaka, 2006, DOI: 10.1017/S0950268805005674; Muangnapoh, 2022, DOI:10.1128/spectrum.00886-22). The discussion for this part should be added for more understanding about the situation facilitating transmission of pathogenic *Vp* by birds. For example, the seasonal factor probably affect in the absence of pathogenic *Vp* in this study as the sample were collected during the cold month(?). Normally, the pathogenic *Vp* favor warmer growth temperature compared to the non-pathogenic *Vp*.

Response: Revised (lines 244-253):

"Also, the bird-carried *V. parahaemolyticus* strains do not carry virulence factors in our study. These *tdh/trh*-negative *V. parahaemolyticus* strains in our study were collected from southeastern China, Guangdong province, in the months of January and March. The *tdh/trh*-positive *V. parahaemolyticus* strains have been reported to be isolated from the mallard feces (Japan in February, the temperature was below 15°C) (8) and brown-headed gull feces (Thailand in March, September, and October, the temperature was above 20°C, according to the Thai Meteorological Department) (18). Therefore, we hypothesized that the acquisition of virulence genes (*tdh/trh*) may be independent of season, temperature, species of migratory birds, and geographical location."

References

8. Miyasaka, J., Yahiro, S., Arahira, Y., Tokunaga, H., Katsuki, K. and Hara-Kudo, Y. (2006) Isolation of *Vibrio parahaemolyticus* and *Vibrio vulnificus* from wild aquatic birds in Japan. *Epidemiol. Infect.*, **134**, 780-785.
18. Muangnapoh, C., Tamboon, E., Supha, N., Toyting, J., Chitrak, A., Kitkumthorn, N., Ekchariyawat, P., Iida, T. and Suthienkul, O. (2022) Multilocus sequence typing and virulence potential of *Vibrio parahaemolyticus* strains isolated from aquatic bird feces. *Microbiol. Spectr.*, **10**, e00886-00822.

6. The template of the references list is inconsistency (e.g. use of capital/small letter, journal abbreviation, italics where needed). It should be revised carefully.

Response: We have corrected the references according to the reviewer's comment.

May 5, 2023

Dr. XueJun Guo

Changchun Veterinary Research Institute, Chinese Academy of Agricultural Sciences/Key laboratory of Jilin Province for Zoonosis Prevention and Control
Changchun
China

Re: Spectrum02170-22R1 (*Vibrio parahaemolyticus* from Migratory birds in China carries an extra copy of tRNA-Gly and plasmid-mediated quinolone resistance (PMQR) gene qnrD)

Dear Dr. XueJun Guo,

Your manuscript has been accepted, and I am forwarding it to the ASM Journals Department for publication. You will be notified when your proofs are ready to be viewed. Please make these two amendments in the proofs:

- 1) Please add the average temperature of the sampling period in China Jan-Mar (line 247) as the figures of those in Japan Thailand were provided. This will support the point that authors bring into the discussion "...acquisition of virulence gene is independent to temperature...".
- 2) Typo error about bird feces strain IDs of isolates from Thailand, they are MUVP(s) not MAVP(s).

Sincerely,

Gaurav Sharma, PhD
Editor, Microbiology Spectrum

Assistant Professor, IIT Hyderabad, India
LabPage: <https://sites.google.com/view/sharmaglab/>
